# Environmental Effects among Differently Located and Fertile Sites on Forest Basal-Area Increment in Temperate Zone

**Pavel Samec** [1,2,*], **Petra Rychtecká** [1], **Miroslav Zeman** [3] **and Miloš Zapletal** [2,4,5]

1 Department of Geology and Soil Science, Faculty of Forestry and Wood Technology, Mendel University, Zemědělská 3, CZ-613 00 Brno, Czech Republic; petra.rychtecka@seznam.cz
2 Global Change Research Institute CAS, Belidla 986/4a, CZ-603 00 Brno, Czech Republic; milos.zapletal@physics.slu.cz
3 Forest Management Institute Brandýs nad Labem, Nábřežní 1326, CZ-250 01 Brandýs nad Labem-Stará Boleslav, Czech Republic; zeman.miroslav@uhul.cz
4 Institute of Physics in Opava, Silesian University in Opava, Bezručovo náměstí 1150/13, CZ-746 01 Opava, Czech Republic
5 Centre for Environment and Land Assessment—Ekotoxa, Otická 37, CZ-746 01 Opava, Czech Republic
* Correspondence: pavel.samec@mendelu.cz

**Abstract:** Environmental properties differently influence the growth of forest tree species. The antagonistic effects of variable environmental properties classify the forest response according to various tree compositions among different sites. The division of the forest response was assessed in 52 stands arranged into 26 types of 13 site management populations (MPs) in 5 areas in the Czech Republic territory. The assessment was performed using time-series multiple regressions of basal-area increment from pure immature stands of Norway spruce (*Picea abies*), Scots pine (*Pinus sylvestris*), oaks (*Quercus* sp.), ash (*Fraxinus excelsior*) and willows (*Salix* sp.) dependent on the interpolated average temperatures, annual precipitation, atmospheric concentrations of $SO_2$, $NO_x$ and $O_3$ and soil properties over the period 1971–2008 at $p < 0.05$. Site MPs differentiated the forest response to a greater extent than tree species. The response of the forests was significantly distributed by means of the montane, upland and waterlogged sites. The multiple determination index ($r^2$) $\geq$ 0.6 indicated an adaptable tree increment but an interval of $r^2$ between 0.80–0.92 implied forest sensitivity to variability in environmental properties on non-waterlogged sites. The index $r^2 < 0.6$ suggested a fluctuating forest increment that reflects environmental variability inconsistently. The fluctuating increment most affected the spruce and pine stands grown from upland to submontane locations. Montane spruce stands, as well as rock pines, appeared to be one of the most sensitive ones to environmental change. Floodplain forests seemed as adaptable to variable environmental properties.

**Keywords:** environmental change; forest ecosystem division; montane spruce forests; natural pines; floodplain forests

## 1. Introduction

Environmental properties differently affect plant growth. While temperatures and water availability directly support plant growth, air pollution disrupts it. However, individual pollutants influence plant growth differently, depending on responses and physiological activity [1]. Plants partially surmount unfavourable environmental properties that do not exceed critical levels by utilising surpluses of favourable environmental properties [2]. The substitution of untoward properties is manifested by similar plant growth simultaneously in favourable and unfavourable conditions. Therefore, the estimation of the combined interaction effects between atmospheric and soil properties indicate ways of living, and community adaptation to environmental change [3].

The environmental change impacts on plant communities are most distinctively regulated by solar activity. Solar radiation and available water are essential to trigger plant

photosynthesis [4]. The effects of insolation alter during solar cycles, but they also vary with the geographical location, time of day and season. Diverse insolation of variously located areas conditions albedo is related to natural organic matter production intensity [5]. However, the albedo decline is caused by both warming and ecosystem transformation. Nevertheless, plant growth is more substantially directed by insolation cycles than by temperatures. The tree species growth is slightly more temperature-dependent within colder seasons than warmer ones [6]. Warmer seasons are accompanied by the far-reaching effects of the atmospheric–oceanic circulation directed by solar activity, while within colder seasons, the distant circulations are replaced by local influences [7].

The compositions of atmosphere and soil prefigure the efficacy of the photosynthetically obtained energy used to create organic matter. The growth processes in plant communities are interconnected with the development of soil carbon content. Soil carbon development simultaneously affects soil fertility and climate variability [8]. The ecosystem carbon cycle transfers the impacts of global environmental changes to the chemical composition of organic matter due to the influence of solar activity cycles on the isotopic composition of plant bodies. The period of the reduced solar activity is manifested by an increase in $^{14}$C isotope content in plants, and vice versa [9]. Carbon interconnects the ecosystem processes by water retention and microbial interactions that reduce C/N during nitrogen access [10]. The soil C/N, bulk density ($D_d$) of individual soil groups, as well as a plant stand density, divide $CO_2$ flow most. At the same time, the development of soil pH is immediately followed by alterations in the Ca/Al, Ca/Mg and Ca/Mn ratios of the wood. Decreasing soil $Ca^{2+}$ availability enhances the $Al^{3+}$ content of the wood, which reduces the Ca/Al ratio [11]. Similarly, decreasing soil pH reduces the Mg uptake by plants, therefore, the Ca/Mg ratio is significantly increased [12]. However, soil C/N and $D_d$ control up to 87% of the dispersion in the $CO_2$ flow from forests [13].

The direct effects of solar radiation on plant growth have been obscured by a cultural restructuring of ecosystems, the intensification of the greenhouse effect and the eutrophication of the environment. The restructuring of the terrestrial ecosystems has had a heavy impact on forests by fragmentation and simplification of the species composition [14]. The diminution in the Earth's surface albedo with the retreat of montane glaciation or deforestation and greenhouse gas emissions accelerated global warming at the end of the Little Ice Age after 1850 [15]. Whereas the retreat of glaciation was naturally caused by the intensification of the solar activity and the rise in summer temperatures, other reasons for the albedo decrease ensued the increasing need of the human race to utilise landscape [16]. Current global warming is unprecedented throughout the Holocene due to human activity [17]. The present-day environmental change has accelerated forest tree species' growth more than the previous Medieval Climatic Optimum [18]. Differences in the forest tree species' growth between colder and warmer areas have decreased [19]. Nevertheless, environmental change does not cause a general improvement in growth conditions, but the overwhelming majority of nutrient-rich sites are threatened by erosion, drainage and acidification [20].

The threat to growth conditions affects not only the forest growth intensity but also causes an imbalance in plant nutrition. Fragmentation and transformation of the forest species' composition have increased soil organic matter losses and $CO_2$ emissions. The subsequent decrease in the return of nutrients from plant shedding to the soil elicits an ecosystem stability loss to mitigate the acidification process [2]. The course of forest ecosystem acidification differs after exceeding critical loads by acid deposition or owing to reduced water availability. The acid deposition load gradually disrupts the ecosystem. The disturbance initially impacts the soil substance interchange until available base cations neutralising the incoming acids are depleted. Only then, does the forest status deteriorate [21].

Restoration of reduced soil fertility as well as adaptation to global warming has become dependent on sufficient soil nitrogen supply to maintain optimally low C/N [22]. Eutrophiering growth of atmospheric $CO_2$ content and physiologically active nitrogen compounds increases a forest increment merely in the conditions of soil nutrient excess and

subdued acidification. Otherwise, eutrophication causes an imbalance between individual substance sources, reducing plant growth [23]. Nevertheless, the growth of the main management tree species at nutrient-rich sites has influenced an increase in the overall increment of European forests since the end of the Little Ice Age [3]. The diameter increment of boreal forests increases by 25% at optimal soil conditions under $2 \times CO_2$ atmosphere [24].

The aim of the presented analysis of variable growth condition effects on the tree species diameter increment was to assess forests' responses to environmental change among different sites. Usual observations of effects on tree species growth were focused either on one tree species in different conditions e.g., [6,11,23,25] or on several tree species under the same conditions [4,25]. The fundamental methods of the tree species growth assessment do not allow for the evaluation of responses in mixed stands to differently acting environmental properties. Variable atmospheric or soil properties influence forest tree species diversely. A diverse effect is indicated by distinctive parameters at correlation functions [8]. Similar correlation parameters suggest forest resilience under various conditions. Nevertheless, the distribution and structure of planted forests prone to acidification or living pests complicate the assessment of the impacts of alterations in growth conditions [23]. The evaluation of the development in unevenly distributed forests became dependent on the growth condition properties' homogenisation. Homogenisation consists of determining the mean courses of variable properties in generalised growth conditions [26]. The generalisation process is based on defining a repeatable series of the landscape geographical division within the unrepeatable area. Each unique area is characterised by a uniform macroclimate; therefore, its structure develops alongside types of relief and rocks [27]. Although altitudinal transitions divide forest growth conditions most significantly, most rock types considerably predetermine soil properties [28]. Within the generalised division of growth conditions, the resilience was assessed between natural and planted stands in the different locations and fertile series. The analysis of the forest growth conditions included both a comparison of various tree species' increments at the same site series and a comparison of the increment of one tree species among various site series [29]. The comparison among different tree species suggests adaptability to the site, while a comparison of various sites suggests a dispersion in adaptation ability.

## 2. Material and Methods

### 2.1. Geographical Pattern

Growth condition effects on the diameter increment of pure forest stands were assessed in terms of atmospheric and soil properties. The growth conditions were characterised by the selection of dominant sites defining the natural environment of the Czech Republic in the Central European landscape. The Czech Republic is situated in a hinterland of transitional temperate climate (78,866 km$^2$; 115–1602 m a.s.l.; 2.0–9.4 °C; 250–1470 mm). The territory of the CR consists mainly of the Bohemian Massif (84.6%) which is covered by the Outer Western Carpathians in the east (15.4%). The Bohemian Massif differs from other Central European montane systems by means of a central basin permeated by rock cities and volcanic mountain ranges [30].

The natural forest cover of the CR exceeds 98%, with the exception of high-mountain, rocky or peatland treeless areas. The current forest cover is 34%. The natural representation of tree species consisted mainly of European beech (*Fagus sylvatica*) (40%), oaks (*Quercus* sp.) (18%), Silver fir (*Abies alba*) (16%), Norway spruce (*Picea abies*) (15%) and Scots pine (*Pinus sylvestris*) (3%) [31], while the current forest composition is mainly composed by 44% of Norway spruce, almost 10% of Scots pine, more than 10% of European beech and 8% of oaks [32].

Forest growth conditions in the Czech Republic are divided into 14 unique areas and 27 management populations (MP) of forest types [33]. Each MP is characterised by the altitudinal zone of climatic conditions and by soil series of acidity and water availability. For the environmental change impact analysis, the forest stands were selected at MPs dominating the Outer Western Carpathians (N), the Eastern Sudetes Mountains (K), the

Elbe River basin (H), the North Bohemian Sandstone Highlands (I) and the volcanic mountain ranges (B). Furthermore, the Outer Western Carpathians are a flysch highland, where mountainous areas are delimited by the Moravian-Silesian Beskids, the Eastern Sudetes Mountains are mostly formed by the crystalline mountains of Hrubý Jeseník and Králický Sněžník. The North Bohemian Sandstone Highlands and the volcanic mountain ranges are not significantly different in altitude, but they contrast with fertility (Figure 1).

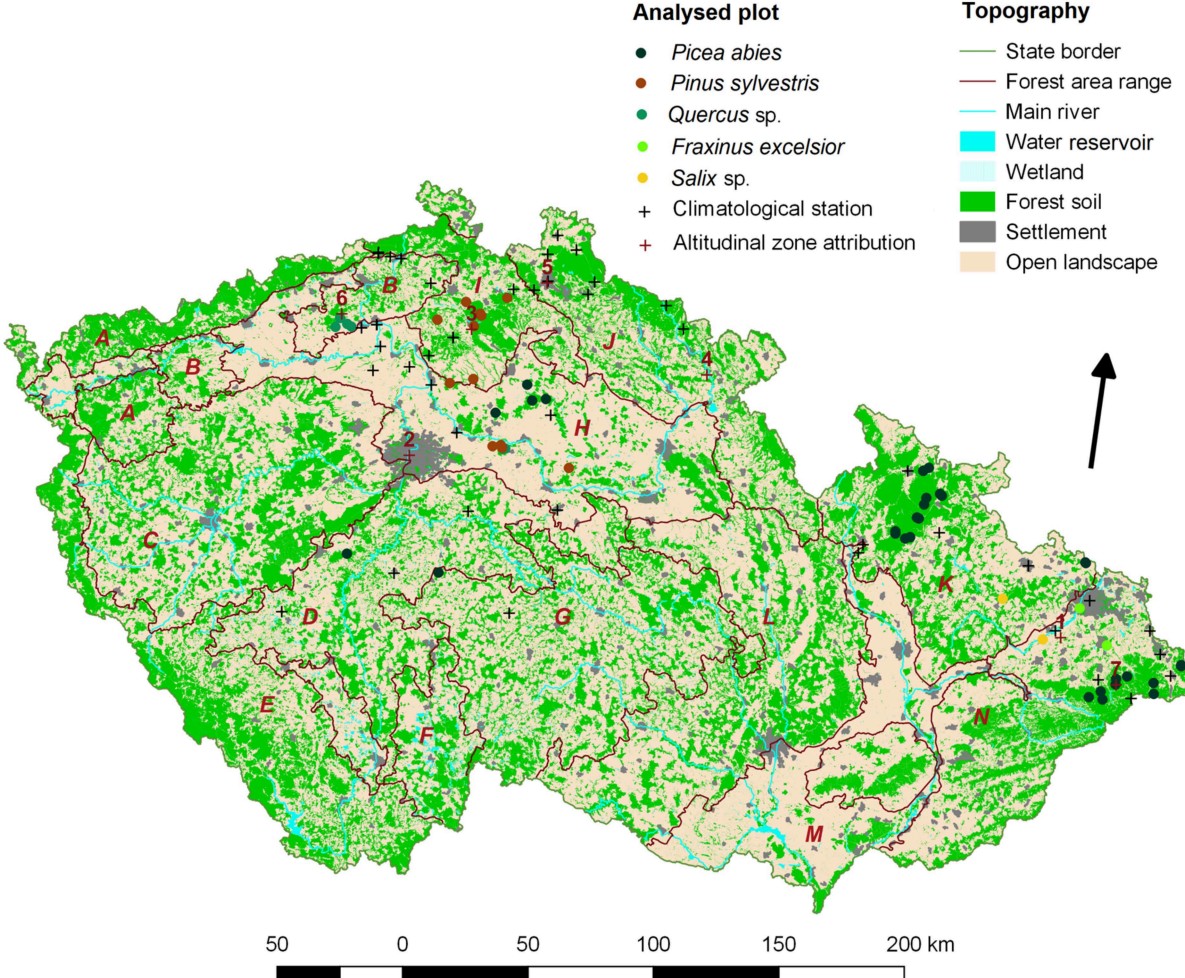

**Figure 1.** Distribution of selected forest stand types in relation to climatological stations including attribution of time series at investigated altitudinal zones. Forest area ranges: A—Ore Mts.; B—Volcanic Mts.; C—Beroun; D—Central Bohemia; E—Bohemian Forest; F—South-Bohemian basins; G—Bohemian-Moravian Highland; H—Elbe basin; I—North-Bohemian sandstone; J—Western Sudetes; K—Eastern Sudetes; L—Moravian Foothills; M—Moravian Floodplains; N—Outer Western Carpathians.

The selection of the forest stands was performed in the MPs, covering >5% of the total forest soil area in each characteristic area. The stands were of managementally significant Norway spruce, Scots pine and oaks, as well as the azonal ecosystems of European ash (*Fraxinus excelsior*) and willows (*Salix* sp.) surviving Quaternary climate changes [34]. Zonal MPs were represented by the site-optimal stand-forming tree species and compared with the managementally transformed composition. Optimal stand-forming tree species for the individual MPs were determined from estimates of potential natural vegetation in the database of the Regional Forest Development Plans (RFDP) [31]. In each MP, the individual stand types were sampled in pairs, allowing for the minimised calculation of statistically representative averages and deviations [35]. The selection was limited to the

single-stage closed stands older than 40 years with a representation of the investigated tree species > 80% within an area > 1 ha [36].

### 2.2. Field Sampling

The selected 52 forest stands were surveyed at the representative plots 30 × 30 m in May–June 2009. The representative plot was located in the geometric centre of the stand. The survey consisted of soil and dendrochronological sampling. An altitude of the surveyed plots was determined as the average through an overlay between vectors of the forest stand representative part and the contour lines spaced 1 m apart in the Basic Base of Geographical Data managed by the Czech Surveyor and Cadastral Administration [37]. The bedrock was typed according to the biogeographical division in the vector model 1:50,000 managed by the Nature Conservation Agency of the Czech Republic [38], while the soil groups were evaluated according to WRB-ISSS-ISRIC [39].

The soil survey included the determination of clay content, $pH/H_2O$, base saturation (BS), $C_{org}$, $Al_2O_3$ and CaO in the upper horizons from five regularly located trenches in each plot. Clay content was determined sedimentographically; pH, acidometrically; BS as the proportion of the base cation contents from the total amount of exchangeable cations by extraction in 0.1M $BaCl_2$ at pH 7.8; $C_{org}$ was determined by infrared spectrophotometry; while the $Al_2O_3$ and CaO contents were determined by AAS from extraction in aqua regia [40].

A dendrochronological sampling of the representative area commenced with the selection of 30 sample trees with the dimensions of the mean stem with a deviation of the breast-height diameter ($d_{1.3}$) ± 2 cm. The tree-ring series were sampled by Pressler borer with a 5-mm inner diameter at the measurement height $d_{1.3}$ in a direction parallel to the slope, avoiding reaction wood [41]. The age, mean diameter, height (*h*), volume ($V_{ha}$) and canopy of the monitored stands were determined by the forest management plans.

### 2.3. Time-Series Analysis

The response of each forest stand was assessed by comparison between the time series of current basal-area increment (BAI) and the series of annual average temperatures (*T*), annual precipitation (*P*), atmospheric concentrations of $SO_2$, $NO_x$ and $O_3$ and soil properties in the period of 1971–2008 (Figure 2). The period length was adapted to the commencement of the assessment of instrumentally measured atmospheric pollution effects on immature forests during climate change [34,42,43]. BAI is a generalisation of the diameter increment which characterises stand density at scale, indicating the effects from the external environment [23]. Climatic properties were interpolated using modified IDW from a rural measurement station system in investigated forest areas [44]. Concentrations of air pollutants were interpolated using ordinary kriging from the Information System on Air Quality of the Czech Hydrometeorological Institute stations [45]. Pollution concentrations before instrumental measuring commencement were derived linearly according to estimations of pre-industrial biogeochemistry [46].

Atmospheric $SO_2$ concentrations were the most serious environmental load in Central Europe until 1994 [47]. Impacts of $NO_x$ and $O_3$ loads worsened significantly, in particular after 2000, due to the intensification of climate change [1,48]. The selection of soil properties from sets of physical and chemical characteristics focused on the attributes correlating with forest health statuses. While pH and BS are fundamental physico-chemical properties indicating acidification and related changes in fertility, the soil clay content forms a matrix maintaining a constant level of base saturation, however, $C_{org}$ affects soil sorption variability [49]. $Al_2O_3$ and CaO contents are components of weathering that are adversely released or bound during acidification [26]. The development of soil properties was generalised by average values characterising particular MPs within selected forest areas in the RFDP database (Figure 3).

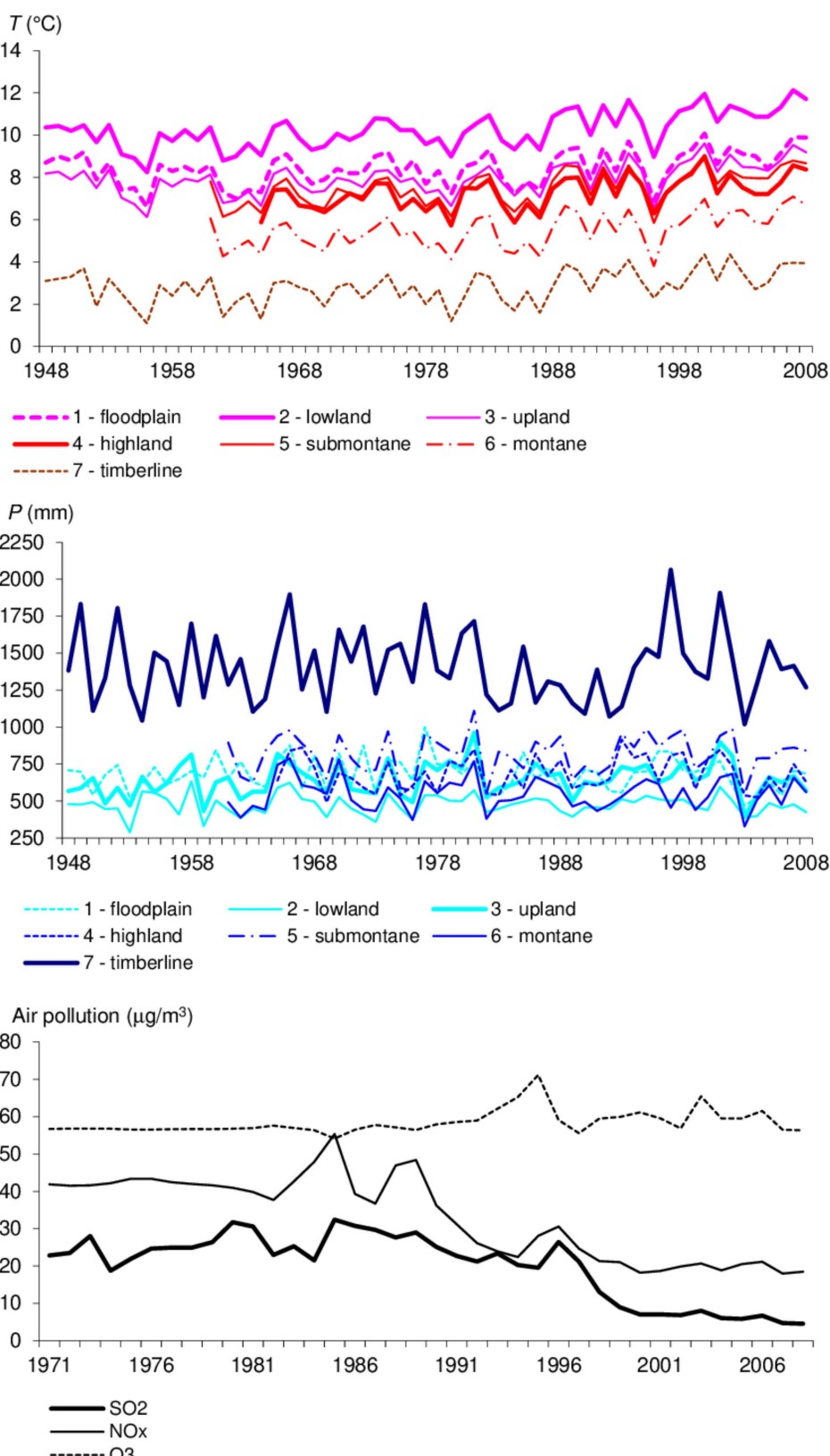

**Figure 2.** Development of year average temperatures (*T*), annual precipitation (*P*) and total atmospheric pollution concentrations at altitudinal zones covering sampled forest stand types.

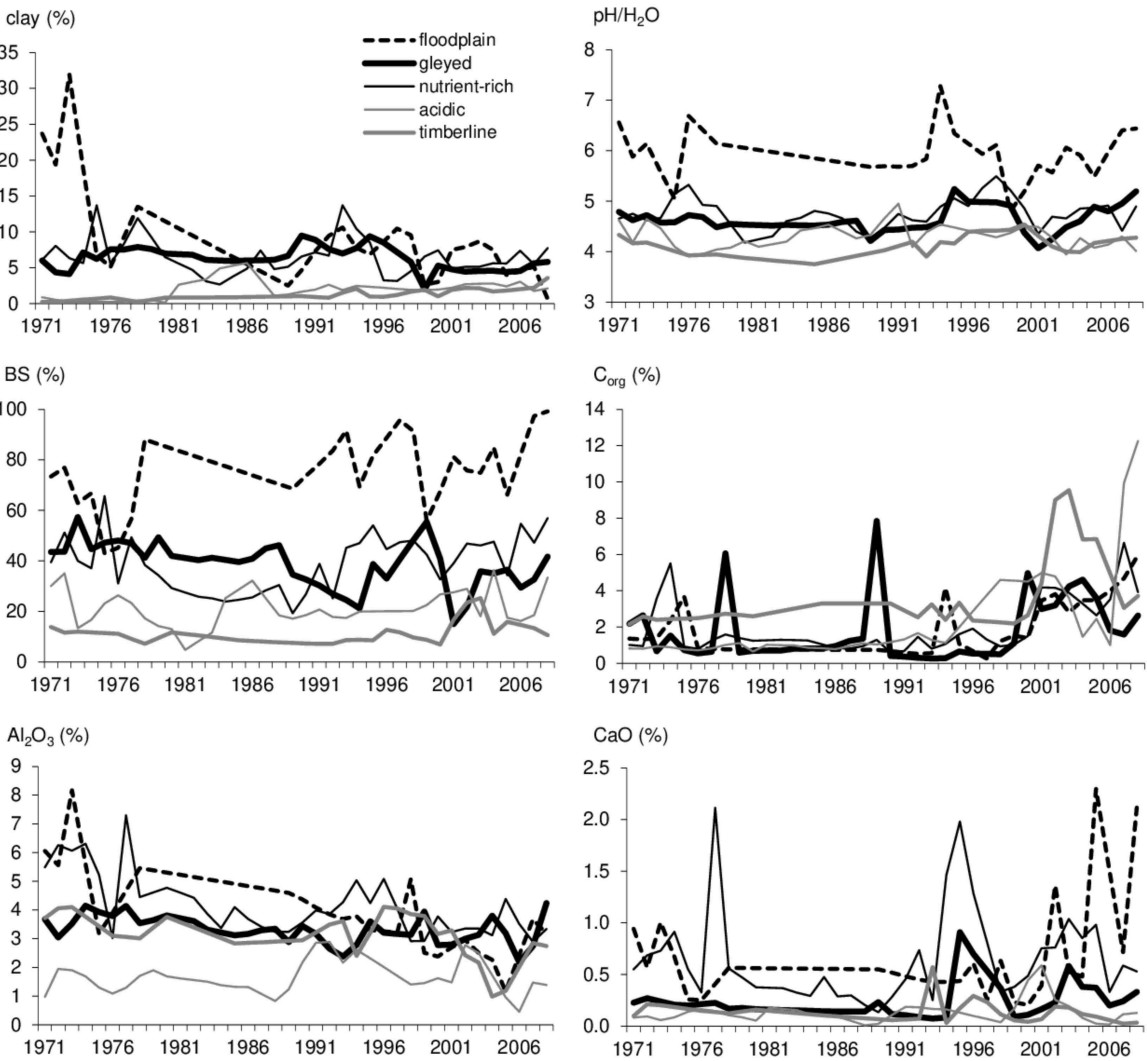

**Figure 3.** Development of soil series characteristics clay content, pH/$H_2O$, base saturation (BS), organic carbon ($C_{org}$) and total $A_2O_3$ and CaO contents.

BAI was obtained from the mean inter-year difference between ratios of standardised stand canopy and basal area ($G_{tab}$) modelled at yield tables [50]. The development of the canopy was characterised by the ratios between growing $V_{ha}$ and modelled $V_{tab}$. Previous $V_{ha}$ were derived from ratios of modelled tree number development and mean stem volume through nonlinear models of dependences between tree-ring increments ($d_{1.3}$) and stand height growth [51]. The $d_{1.3}$ widths were measured using a VIAS TimeTable measuring system (SCIEM) with an accuracy of 10 μM. Values of $d_{1.3}$ were synchronised visually, cross-dated by PAST 4 [52] and controlled statistically using COFECHA [53].

*2.4. Statistical Assessment*

The statistical assessment of forest response was divided into exploratory analysis and multiple regressions. The exploratory analysis consisted of linear correlations and residue analysis. While linear correlations indicated the effects of the individual growth condition properties on stand type BAI, the residue analysis was designed to reveal inaccuracies in the linear models. Linear correlations were evaluated as significant at $p < 0.05$, less significant between $p \geq 0.05$ and $p < 0.50$, and insignificant. Residue analysis was performed using the residual standard deviation (RSD), elevation ($E_R$) and skewness ($A_R$). The residues were calculated as differences between obtained and modelled BAIs. The low linear approximation was detected in $E_R > RSD^4$ or $A_R > RSD^3$ [54]. Multiple regressions were

calculated for each plot, but merely the parameters of the most relevant function from a pair of the same stand types were considered at $p < 0.05$. Independent environmental variables were normalised by z-transformation expressing deviations of the measured values from the mean. The multiple model was compiled as follows:

$$\text{BAI} = \sum_{i=1}^{m} a_i . x_i + b \tag{1}$$

where $x_i$ is the z-transformed growth condition property, $a_i$ is the slope parameter of the growth condition property, $b$ is the intercept parameter, $i$ is the sequence of the growth condition property and $m$ is the number of the growth condition properties. Functions among the individual stand types were subsequently contrasted using minimal Fischer–Snedecor criteria ($F_{\min}$) and determination indices ($r^2$) [55].

## 3. Results

### 3.1. Management Populations of Forest Growth Conditions

Management populations of forest soils are unevenly represented in the individual areas of the Czech Republic. Simultaneously, no MP is dominant in all the forest areas of the Czech Republic. Only six MPs exceeded their representation of 5% of the forest soil area. The six predominant MPs include 62.9% of the forest soils in the Czech Republic. The acidic sites predominate from the uplands to the mountains, while the nutrient-rich sites are only from the highlands to the mountains. The upland acidic series includes 5.7% of the forest soils, highlands (10.7%) and montane (11.5%). The nutrient-rich series includes 17.9% in the highlands while merely 11.1% in the montane locations. The montane sites are characteristic of a larger proportion of the gleyed series (5.9%) (Table 1). The upland acidic sites are most concentrated in the area of the Moravian floodplains (M). The highland acidic sites are most concentrated in the Beroun areas (C) and the Central Bohemian areas (D) while the highland nutrient-rich sites are most concentrated separately in the area of the volcanic mountain ranges (B) and further surroundings of the Moravian floodplains in the Bohemian-Moravian intermountain (L) and Eastern Sudetes (K), to the Carpathian (N) areas. The montane acidic sites predominate in the largest mountain ranges in the territory of the Czech Republic from the Ore Mountains (A), the Šumava Range (E), and the Bohemian-Moravian Highlands (G) to the Western Sudetes (J). The montane nutrient-rich to gleyed sites do not dominate in any of the Czech forest areas, but mostly, except for the gleyed series in the Eastern Sudetes and the Carpathians, they come after the dominant soil series representation.

Selected forest areas demonstrating the diversity of the Czech Republic are specific for four dominant site MPs. Forest soils in the selected areas are most represented by upland (9.5%), highland (28.1%) montane (16.8%) and nutrient-rich sites and rock towns (8.1%). Highland nutrient-rich sites dominate in three of the five selected areas. They cover 28% of the forests in the volcanic mountain ranges but more than 44% of the forests in the Outer Western Carpathians. Volcanic mountain ranges are typical of the common dominance of the upland (16.4%) to highland (28.0%) nutrient-rich sites, while the Elbe River basin (H) is mainly formed by acidic (21.7%) to nutrient-rich (34.7%) sites in the uplands. Contrarily, the Eastern Sudetes and the Outer Western Carpathians are similarly characterised by a predominance of nutrient-rich series from the highlands (34.7% and 44.2%) to montane altitudes (25.0% and 23.5%). The included North Bohemian Sandstone Highlands (I) differ, mainly in the predominance of rock cities (40.3%) followed by highland acidic soils (14.7%).

**Table 1.** Proportion of forest type management populations composed of relief types and soil series in forest area ranges of the Czech Republic (%). For an explanation of forest area ranges locations see Figure 1.

| Relief | Series | A | B | C | D | E | F | G | H | I | J | K | L | M | N | CR |
|---|---|---|---|---|---|---|---|---|---|---|---|---|---|---|---|---|
| Floodplain | Riverine | 0.2 | 0.1 | 0.1 | 0.7 | 0.1 | 1.1 | 0.0 | 4.2 | 0.4 | 0.1 | 1.0 | 0.5 | 44.1 | 2.0 | 1.5 |
| | Swamp | 1.1 | 1.6 | 1.1 | 0.9 | 0.7 | 1.0 | 1.0 | 0.9 | 0.5 | 0.6 | 1.3 | 0.5 | 0.9 | 0.0 | 0.8 |
| Rocky | Pine | 0.7 | 0.4 | 7.8 | 0.4 | 0.1 | 17.7 | 0.2 | 15.3 | 40.3 | 1.2 | - | 0.4 | 0.4 | - | 4.0 |
| Upland | Exposed | 0.1 | 7.8 | 5.1 | 4.2 | 0.1 | 0.3 | 0.0 | 2.4 | 2.4 | 0.1 | 0.0 | 4.4 | 0.7 | 0.3 | 1.8 |
| | Acidic | 1.1 | 3.4 | 14.3 | 10.9 | 0.4 | 3.2 | 0.1 | 21.7 | 5.2 | 0.7 | 0.4 | 11.5 | 30.3 | 0.5 | 5.7 |
| | Nutrient-rich | 0.0 | 16.4 | 4.5 | 4.1 | 0.0 | 1.2 | 0.0 | 34.7 | 2.4 | 1.8 | 0.3 | 6.1 | 21.0 | 9.5 | 4.8 |
| | Gleyed | 0.6 | 0.0 | 2.9 | 0.7 | 0.1 | 10.4 | 0.2 | 17.7 | 0.3 | 0.1 | 0.1 | 0.2 | 0.8 | 0.7 | 1.6 |
| Karstic | Exposed | 0.2 | 3.3 | 0.1 | 0.7 | 0.1 | 0.0 | 0.2 | - | - | - | 0.5 | 1.2 | - | 0.1 | 0.4 |
| | Nutrient-rich | - | - | - | 0.0 | 0.1 | 0.0 | 0.0 | - | - | - | 0.0 | 0.6 | - | - | 0.1 |
| | Wet | 0.0 | - | 0.0 | 0.0 | 0.0 | 6.6 | 0.1 | - | 0.5 | - | - | 0.0 | - | - | 0.2 |
| Foothill/highland | Ravine | 3.0 | 4.6 | 2.5 | 1.9 | 3.0 | 3.4 | 0.5 | 0.6 | 6.8 | 2.3 | 1.2 | 2.1 | 1.1 | 0.6 | 2.1 |
| | Exposed | 1.0 | 10.4 | 4.4 | 5.3 | 1.3 | 0.8 | 1.6 | 0.0 | 2.9 | 3.1 | 6.1 | 8.0 | 0.0 | 4.4 | 3.7 |
| | Acidic | 7.1 | 5.7 | 22.5 | 31.1 | 5.8 | 9.1 | 7.2 | 1.4 | 14.7 | 14.7 | 2.7 | 12.7 | - | 1.3 | 10.7 |
| | Nutrient-rich | 0.2 | 28.0 | 13.4 | 24.2 | 3.9 | 5.4 | 6.5 | 0.5 | 4.1 | 8.6 | 34.7 | 40.5 | 0.7 | 44.2 | 17.9 |
| | Gleyed | 0.3 | 3.5 | 11.2 | 14.2 | 1.4 | 22.7 | 3.4 | 0.1 | 2.4 | 1.3 | 2.3 | 2.2 | 0.1 | 2.0 | 4.4 |
| Montane | Exposed | 9.9 | 3.1 | 0.9 | 0.0 | 7.0 | - | 3.5 | - | 1.0 | 8.6 | 6.5 | 0.7 | - | 9.7 | 4.4 |
| | Acidic | 28.1 | 0.2 | 2.2 | 0.0 | 29.0 | 0.0 | 29.7 | - | 9.5 | 21.9 | 5.6 | 2.8 | - | 0.4 | 11.5 |
| | Nutrient-rich | 10.6 | 9.1 | 0.5 | 0.2 | 12.8 | 0.1 | 22.7 | - | 3.7 | 15.2 | 25.0 | 3.2 | - | 23.5 | 11.1 |
| | Gleyed | 7.6 | 2.0 | 5.4 | 0.1 | 14.5 | 0.1 | 20.2 | - | 1.6 | 3.9 | 4.1 | 2.0 | - | 0.4 | 5.9 |
| | Wet | 0.5 | 0.3 | 0.9 | 0.4 | 2.1 | 17.0 | 1.9 | 0.3 | 1.4 | 1.0 | 0.6 | 0.3 | - | 0.1 | 1.3 |
| High-mountain | Exposed | 0.7 | - | - | - | 0.9 | - | - | - | - | 2.1 | 1.3 | - | - | 0.1 | 0.5 |
| | Acidic | 15.2 | - | - | - | 5.6 | - | - | - | - | 5.3 | 1.7 | - | - | 0.0 | 2.2 |
| | Nutrient-rich | 0.4 | - | - | - | 1.1 | - | - | - | - | 0.1 | 1.6 | - | - | 0.1 | 0.3 |
| | Gleyed | 1.9 | - | - | - | 3.9 | - | 0.1 | - | - | 0.7 | 0.3 | - | - | 0.0 | 0.7 |
| | Wet | 9.1 | - | 0.2 | - | 3.7 | - | 0.9 | - | - | 1.7 | 0.1 | - | - | 0.0 | 1.3 |
| | Timberline | 0.4 | - | - | - | 2.1 | - | - | - | - | 3.5 | 2.2 | - | - | 0.0 | 0.8 |
| | Supalpine | 0.0 | - | - | - | - | - | - | - | - | 1.5 | 0.3 | - | - | - | 0.1 |

*3.2. Properties of Assessed Ecosystems*

3.2.1. Growth Conditions

The survey was conducted in 26 forest stand types at 13 MPs of four altitudinal zones and 11 soil series. The soil series was composed of 11 bedrock types and 15 soil groups. The selected forest stand types were divided in parallel between the Bohemian Massif and the Outer Western Carpathians. Conditions of the Bohemian Massif were characterised by 10 stand types in the upland areas, 7 stand types in the montane areas, and 2 in the rock cities. The Western Carpathians were defined by 2 types of floodplain forests and 5 types of montane forest stands. The montane sites were similarly defined by foothill to ravine, exposed, nutrient-rich and the peak timberline series, whereas the uplands were evaluated by means of the correlation between flat, exposed, nutrient-rich and the gleyed series.

The selected forest stands were located in the range of altitudes between 192–1173 m. The lowest-lying one was the pine stand in the upland gleyed series. The highest-lying one was the montane spruce stand in the Outer Western Carpathians. The extreme values of the growth conditions intervened several times in floodplain willow beds, gleyed upland pine and oak stands and montane spruce forests. While the lowest temperatures of +3.51 °C occurred at the sites of the Sudeten montane spruce forests; the lowest $SO_2$ and $NO_x$ concentrations and the highest $O_3$ load appeared in the Carpathian montane spruce forests. On the contrary, the gleyed upland sites of the pine and oak stands were most burdened by $SO_2$ and $NO_x$ atmospheric pollution, however, only by minimal concentrations of tropospheric $O_3$. Similarly, the lowest annual precipitation of 550 mm was detected in the nutrient-rich sites of the upland oak stands while the highest precipitation of more than 1280 mm accompanied the Carpathian ravine sites (Table 2).

The marginal values of the soil properties most often accompanied the nutrient-rich sites in contrast to the rock cities. The most frequent bedrock of the selected forests was formed by sandstones. Sandstones centrically occurred in the Carpathian forests and in the North Bohemian steep reliefs. In this context, Arenic Podzols and Skeletic Cambisols were more frequent than Haplic Cambisols, concentrated more on igneous rocks. However, Luvisols, Luvic Cambisols and Entic Podzols associations, which are predominant in the inland uplands, were similarly common. The lowest soil clay content of 1.1% was discovered in the exposed Sudetes sites of the spruce forests, while the highest soil clay content of 31.5% was determined at the nutrient-rich sites of the upland areas in oak stands. The lowest soil $pH/H_2O$ 3.7 was ascertained at the Carpathian nutrient-rich sites while the highest $pH/H_2O$ 6.1 was discovered in the submontane floodplains. Nevertheless, the highest pH corresponded to the highest soil base saturation at the floodplain sites (97.7%), whereas the lowest BS (11.3%) was identified at the foot of the Outer Western Carpathians. The values of the soils' chemical properties followed the distribution of bedrock types rather than relief, such as the pH or BS. The lowest $C_{org}$ content was demonstrated in the natural pines from the rock city areas (0.75%) while the highest $C_{org}$ content was determined in the Carpathian montane spruce forests (17%). Similarly, the lowest $Al_2O_3$ content was discovered in the soils from the rock cities (only 0.07%) but the highest content of total aluminium was identified in the nutrient-rich sites of the Elbe oak stands (5.6%). On the contrary, the lowest soil CaO contents of 0.01% partially corresponded to the occurrence of the lowest pH and BS in the Outer Western Carpathians, but the highest CaO content was recorded in the Elbe exposed oak groves (2.09%) (Table 3).

**Table 2.** Air condition characteristics of sampled forest stand types (mean ± standard deviation).

| Geotectonics | Relief | Series | Tree-Species | Origin | Altitude | $T$ | $P$ | $SO_2$ | $NO_x$ | $O_3$ |
|---|---|---|---|---|---|---|---|---|---|---|
| Bohemian Massif | Rocky | Pine | *Pinus sylvestris* | Natural | 280–330 | 7.52 ± 0.50 | 687 ± 50 | 6.67 ± 1.43 | 21.03 ± 1.20 | 54.42 ± 2.07 |
| | | | *Pinus sylvestris* | Planted | 265–296 | 7.59 ± 0.21 | 648 ± 3 | 6.72 ± 1.46 | 21.43 ± 1.22 | 53.65 ± 2.02 |
| | Upland | Flat | *Picea abies* | Planted | 255–293 | 8.29 ± 0.13 | 690 ± 10 | 6.75 ± 1.48 | 21.74 ± 1.25 | 53.03 ± 1.99 |
| | | Exposed | *Quercus petraea* | Natural | 516–588 | 7.04 ± 0.16 | 582 ± 14 | 6.19 ± 1.18 | 17.04 ± 1.00 | 62.26 ± 2.74 |
| | | | *Pinus sylvestris* | Planted | 245–267 | 8.33 ± 0.05 | 559 ± 44 | 6.77 ± 1.48 | 21.83 ± 1.25 | 52.86 ± 1.98 |
| | | | *Picea abies* | Planted | 457–503 | 7.17 ± 0.28 | 643 ± 35 | 6.33 ± 1.24 | 18.20 ± 1.04 | 59.97 ± 2.52 |
| | | Nutrient-rich | *Quercus petraea* | Natural | 386–398 | 7.69 ± 0.04 | 550 ± 4 | 6.50 ± 1.33 | 19.63 ± 1.11 | 57.18 ± 2.27 |
| | | | *Pinus sylvestris* | Planted | 197–262 | 8.56 ± 0.02 | 578 ± 5 | 6.82 ± 1.52 | 22.26 ± 1.29 | 52.02 ± 1.94 |
| | | | *Picea abies* | Planted | 236–253 | 8.47 ± 0.01 | 572 ± 9 | 6.79 ± 1.50 | 22.02 ± 1.27 | 52.49 ± 1.96 |
| | | Gleyded | *Quercus robur* | Natural | 195–200 | 8.58 ± 0.01 | 559 ± 2 | 6.88 ± 1.56 | 22.78 ± 1.33 | 51.00 ± 1.89 |
| | | | *Pinus sylvestris* | Planted | 192–203 | 8.59 ± 0.06 | 556 ± 7 | 6.88 ± 1.56 | 22.78 ± 1.33 | 51.00 ± 1.89 |
| | | | *Picea abies* | Planted | 277–319 | 8.33 ± 0.29 | 570 ± 6 | 6.68 ± 1.44 | 21.15 ± 1.20 | 54.20 ± 2.06 |
| | Montane | Foothill | *Picea abies* | Planted | 500–704 | 6.49 ± 0.61 | 940 ± 65 | 6.15 ± 1.17 | 16.70 ± 1.00 | 62.91 ± 2.82 |
| | | Ravine | *Pinus sylvestris* | Natural | 321–371 | 6.88 ± 0.63 | 624 ± 44 | 6.36 ± 1.27 | 18.46 ± 1.07 | 59.47 ± 2.50 |
| | | Exposed | *Picea abies* | Planted | 805–840 | 5.05 ± 0.31 | 976 ± 63 | 5.68 ± 1.02 | 12.84 ± 1.01 | 70.49 ± 3.65 |
| | | Nutrient-rich | *Picea abies* | Planted | 741–820 | 5.17 ± 0.43 | 963 ± 56 | 5.72 ± 1.03 | 13.12 ± 1.00 | 69.93 ± 3.58 |
| | | Gleyed | *Picea abies* | Planted | 800–840 | 4.99 ± 0.16 | 873 ± 5 | 5.66 ± 1.01 | 12.70 ± 1.01 | 70.76 ± 3.68 |
| | | Wet | *Picea abies* | Planted | 849–860 | 4.82 ± 0.04 | 871 ± 2 | 5.60 ± 1.00 | 12.14 ± 1.03 | 71.86 ± 3.81 |
| | | Timberline | *Picea abies* | Natural | 1046–1123 | 3.51 ± 0.34 | 1005 ± 21 | 5.15 ± 1.02 | 8.42 ± 1.26 | 79.16 ± 4.70 |
| Carpathians | Floodplain | Swamp | *Salix fragilis* | Natural | 235–335 | 7.90 ± 0.40 | 677 ± 13 | 6.71 ± 1.45 | 21.36 ± 1.22 | 53.79 ± 2.04 |
| | | Riverine | *Fraxinus excelsior* | Natural | 223–313 | 7.96 ± 0.37 | 796 ± 145 | 6.68 ± 1.44 | 21.14 ± 1.21 | 54.22 ± 2.07 |
| | Montane | Foothill | *Picea abies* | Planted | 581–626 | 6.61 ± 0.33 | 1107 ± 19 | 6.17 ± 1.17 | 16.88 ± 0.99 | 62.57 ± 2.77 |
| | | Ravine | *Picea abies* | Planted | 1050–1150 | 3.71 ± 0.44 | 1281 ± 88 | 5.12 ± 1.03 | 8.17 ± 1.28 | 79.65 ± 4.76 |
| | | Exposed | *Picea abies* | Planted | 559–811 | 5.83 ± 0.62 | 1160 ± 77 | 5.89 ± 1.07 | 14.54 ± 0.99 | 67.16 ± 3.27 |
| | | Nutrient-rich | *Picea abies* | Planted | 697–1062 | 5.31 ± 1.02 | 1191 ± 82 | 5.65 ± 1.04 | 12.61 ± 1.05 | 70.93 ± 3.71 |
| | | Timberline | *Picea abies* | Natural | 1136–1173 | 3.59 ± 0.19 | 1209 ± 11 | 5.01 ± 1.05 | 7.28 ± 1.34 | 81.39 ± 4.98 |

$T$—temperature (°C); $P$—precipitation (mm); $SO_2$—sulphur dioxide (g/m$^3$); $NO_x$—nitrogen oxides (g/m$^3$); $O_3$—tropospheric ozone (g/m$^3$).

**Table 3.** Soil characteristics of sampled forest stand types in relation to bedrock type (mean ± standard deviation).

| Geotectonics | Relief | Series | Tree-Species | Origin | Bedrock | Soil Group | clay | pH | BS | $C_{org}$ | $Al_2O_3$ | CaO |
|---|---|---|---|---|---|---|---|---|---|---|---|---|
| Bohemian Massif | Rocky | Pine | *Pinus sylvestris* | Natural | sandstone | Arenic Podzol | 1.17 ± 0.66 | 4.19 ± 0.25 | 25.46 ± 14.40 | 0.75 ± 0.58 | 0.07 ± 0.03 | 0.02 ± 0.01 |
| | | | *Pinus sylvestris* | Planted | sandstone | Arenic Podzol | 2.05 ± 1.02 | 4.07 ± 0.26 | 19.14 ± 7.70 | 0.83 ± 0.69 | 0.13 ± 0.07 | 0.02 ± 0.03 |
| | Upland | Flat | *Picea abies* | Planted | loess | Haplic Luvisol | 2.73 ± 2.00 | 3.94 ± 0.24 | 12.65 ± 6.86 | 4.08 ± 1.38 | 1.75 ± 0.15 | 0.11 ± 0.02 |
| | | Exposed | *Quercus petraea* | Natural | phonolite | Haplic Cambisol | 17.97 ± 11.28 | 5.27 ± 0.65 | 70.47 ± 32.88 | 5.61 ± 1.81 | 4.10 ± 0.33 | 2.09 ± 0.25 |
| | | | *Pinus sylvestris* | Planted | sandstone | Luvic Cambisol | 15.44 ± 10.54 | 5.85 ± 1.75 | 60.98 ± 42.31 | 1.22 ± 0.32 | 1.27 ± 0.39 | 1.21 ± 1.92 |
| | | | *Picea abies* | Planted | granite | Haplic Cambisol | 12.79 ± 26.22 | 4.22 ± 0.22 | 18.29 ± 5.56 | 2.46 ± 0.63 | 2.99 ± 0.36 | 0.18 ± 0.11 |
| | | Nutrient-rich | *Quercus petraea* | Natural | phonolite | Haplic Luvisol | 31.47 ± 20.88 | 5.48 ± 0.45 | 92.56 ± 7.09 | 6.46 ± 2.95 | 5.60 ± 1.61 | 0.80 ± 0.22 |
| | | | *Pinus sylvestris* | Planted | phonolite | Epidystric Stagnosol | 2.13 ± 0.35 | 4.34 ± 0.60 | 32.78 ± 19.62 | 2.65 ± 0.83 | 1.03 ± 0.13 | 0.06 ± 0.04 |
| | | | *Picea abies* | Planted | marlstone | Luvic Cambisol | 19.1 ± 15.55 | 3.87 ± 0.11 | 20.63 ± 3.96 | 2.95 ± 2.43 | 1.75 ± 1.33 | 0.04 ± 0.02 |
| | | Gleyded | *Quercus robur* | Natural | gravel-sand | Endogleyic Stagnosol | 9.85 ± 10.47 | 4.35 ± 0.33 | 22.2 ± 21.23 | 1.75 ± 0.44 | 0.72 ± 0.24 | 0.03 ± 0.03 |
| | | | *Pinus sylvestris* | Planted | gravel-sand | Entic Podzol | 4.86 ± 4.75 | 4.46 ± 0.94 | 34.07 ± 32.61 | 1.21 ± 1.06 | 0.61 ± 0.20 | 0.05 ± 0.05 |
| | | | *Picea abies* | Planted | gravel-sand | Stagni-luvic Cambisol | 19.62 ± 23.46 | 3.85 ± 0.10 | 11.37 ± 3.41 | 4.43 ± 2.35 | 2.02 ± 1.92 | 0.05 ± 0.05 |
| | Montane | Foothill | *Picea abies* | Planted | amphibolite | Haplic Cambisol | 2.41 ± 3.22 | 4.40 ± 0.67 | 41.09 ± 32.07 | 9.81 ± 7.17 | 4.16 ± 0.46 | 0.37 ± 0.14 |
| | | Ravine | *Pinus sylvestris* | Natural | sandstone | Arenic Podzol | 6.38 ± 8.09 | 3.72 ± 0.09 | 16.62 ± 6.77 | 1.67 ± 0.70 | 0.33 ± 0.11 | 0.02 ± 0.01 |
| | | Exposed | *Picea abies* | Planted | orthogneiss | Skeletic Cambisol | 1.10 ± 0.54 | 3.91 ± 0.14 | 16.44 ± 2.96 | 8.96 ± 4.06 | 3.35 ± 0.54 | 0.13 ± 0.06 |
| | | Nutrient-rich | *Picea abies* | Planted | phyllite | Entic Podzol | 0.98 ± 0.40 | 4.04 ± 0.31 | 17.65 ± 10.1 | 7.15 ± 2.18 | 3.28 ± 1.24 | 0.15 ± 0.04 |
| | | Gleyed | *Picea abies* | Planted | paragneiss | Stagnic Cambisol | 1.68 ± 0.73 | 3.97 ± 0.32 | 12.05 ± 1.33 | 5.85 ± 1.81 | 3.62 ± 1.19 | 0.16 ± 0.06 |
| | | Wet | *Picea abies* | Planted | paragneiss | Haplic Gleysol | 2.53 ± 0.63 | 4.02 ± 0.20 | 15.5 ± 6.41 | 6.08 ± 1.94 | 4.15 ± 1.31 | 0.14 ± 0.07 |
| | | Timberline | *Picea abies* | Natural | orthogneiss | Haplic Podzol | 2.60 ± 0.90 | 3.86 ± 0.18 | 16.56 ± 5.8 | 9.19 ± 8.11 | 1.25 ± 0.49 | 0.07 ± 0.02 |

**Table 3.** *Cont.*

| Geotectonics | Relief | Series | Tree-Species | Origin | Bedrock | Soil Group | clay | pH | BS | C$_{org}$ | Al$_2$O$_3$ | CaO |
|---|---|---|---|---|---|---|---|---|---|---|---|---|
| Carpathians | Floodplain | Swamp | *Salix fragilis* | Natural | fluvial loam | Stagnic Fluvisol | 16.50 ± 2.15 | 6.10 ± 0.03 | 97.72 ± 2.47 | 4.60 ± 1.46 | 2.29 ± 0.33 | 0.25 ± 0.05 |
| | | Riverine | *Fraxinus excelsior* | Natural | fluvial loam | Haplic Fluvisol | 3.18 ± 2.34 | 5.34 ± 1.06 | 66.77 ± 36.38 | 4.07 ± 1.35 | 3.18 ± 1.07 | 0.39 ± 0.32 |
| | Montane | Foothill | *Picea abies* | Planted | sandstone | Haplic Cambisol | 1.29 ± 0.32 | 3.78 ± 0.08 | 11.34 ± 4.52 | 10.64 ± 5.61 | 2.68 ± 0.29 | 0.08 ± 0.03 |
| | | Ravine | *Picea abies* | Planted | sandstone | Umbric Cambisol | 15.66 ± 8.11 | 3.95 ± 0.13 | 5.40 ± 1.14 | 4.40 ± 1.01 | 2.55 ± 0.42 | 0.01 ± 0.00 |
| | | Exposed | *Picea abies* | Planted | sandstone | Skeletic Cambisol | 1.24 ± 0.72 | 3.81 ± 0.17 | 15.63 ± 7.08 | 13.75 ± 3.46 | 2.65 ± 0.38 | 0.10 ± 0.05 |
| | | Nutrient-rich | *Picea abies* | Planted | sandstone | Skeletic Cambisol | 1.59 ± 0.56 | 3.67 ± 0.07 | 11.63 ± 3.60 | 16.23 ± 5.82 | 2.45 ± 0.40 | 0.06 ± 0.01 |
| | | Timberline | *Picea abies* | Natural | sandstone | Haplic Podzol | 1.22 ± 0.74 | 3.72 ± 0.10 | 17.85 ± 6.46 | 17.02 ± 14.25 | 1.38 ± 0.46 | 0.07 ± 0.05 |

Clay—soil clay (<2 μM) content (%); BS—base saturation (%); C$_{org}$—organic carbon (%); Al$_2$O$_3$—aluminium oxide (%); CaO—calcium oxide (%).

### 3.2.2. Stand Types

The characteristics of the selected stand types were most often distributed along the uplands and mountain ranges. The floodplain forests were characterised by features similar to the submontane stands, although their location was related to the upland sites. Edge stand characteristics values occurred more often in the willow woods, ravine pines and exposed Carpathian forests. The willow stands were the youngest ones at the time of the survey (45 years) while the North Bohemian ravine pines were the oldest ones (123 years). Although most of the surveyed forest stands were younger than 100 years, natural rock pines, exposed upland pines and spruce stands, as well as nutrient-rich oak groves, were older than 100 years.

The low age foreshadowed the lowest ascertained stand height of $14 \pm 8$ m in the willow stands but did not significantly affect the maximum height $> 30$ m in the exposed Carpathian spruce stands. The mean $d_{1.3}$ was smallest in the planted rock pines (22 cm), while on average, the largest $d_{1.3} > 39$ cm was detected in the spruce stands at the foot of the Carpathians. The mean $d_{1.3} > 30$ cm was most frequently discovered in the mountain forests, whereas smaller mean trunk diameters predominated in the upland forests. The stand height affected marginal increment values. The lowest basal-area increment was $<2.5$ m$^2$/ha.year and occurred in stands with a height of $<20$ m. On the contrary, the highest increments $> 8$ m$^2$/ha.year occurred in stands higher than 20 m on rich riverine or foothill sites (Table 4; Figure 4).

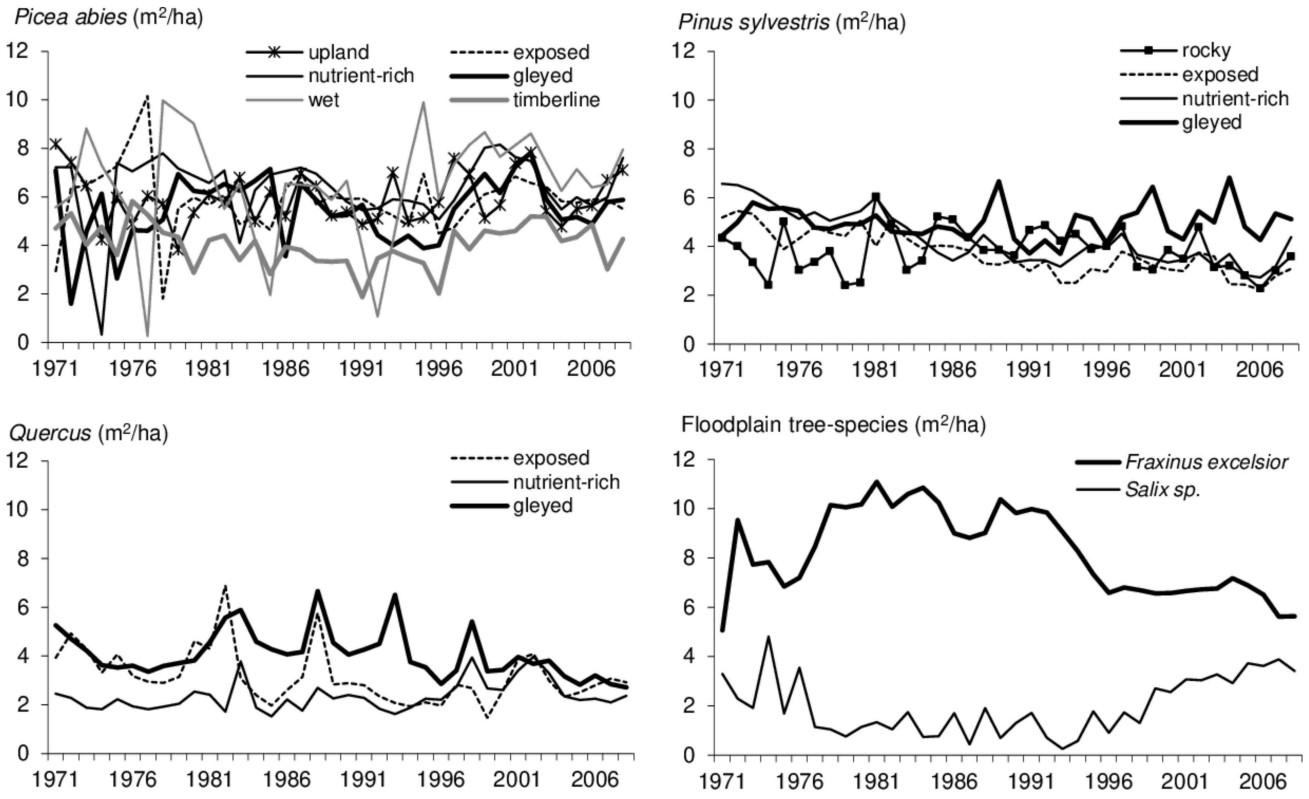

**Figure 4.** Current basal-area increment of forest stands selected on various soil series.

**Table 4.** Grow characteristics of sampled forest stand types (mean ± standard deviation).

| Geotectonics | Relief | Series | Tree-Species | Origin | Age | $V_{ha}$ | h | $d_{1.3}$ | BAI |
|---|---|---|---|---|---|---|---|---|---|
| Bohemian Massif | Rocky | Pine | *Pinus sylvestris* | Natural | 115 ± 2 | 296 ± 4 | 23 ± 1 | 24.00 ± 2.07 | 4.48 ± 1.62 |
| | | | *Pinus sylvestris* | Planted | 89 ± 3 | 225 ± 4 | 19 ± 4 | 22.00 ± 2.02 | 3.79 ± 1.53 |
| | Upland | Flat | *Picea abies* | Planted | 66 ± 13 | 278 ± 54 | 21 ± 10 | 23.00 ± 1.99 | 7.45 ± 4.4 |
| | | Exposed | *Quercus petraea* | Natural | 77 ± 1 | 151 ± 1 | 19 ± 1 | 22.50 ± 2.74 | 3.17 ± 1.30 |
| | | | *Pinus sylvestris* | Planted | 103 ± 1 | 256 ± 2 | 21 ± 4 | 25.00 ± 1.98 | 3.72 ± 1.37 |
| | | | *Picea abies* | Planted | 109 ± 1 | 342 ± 26 | 23 ± 1 | 24.50 ± 2.52 | 5.28 ± 1.62 |
| | | Nutrient-rich | *Quercus petraea* | Natural | 100 ± 2 | 203 ± 25 | 19 ± 3 | 27.00 ± 2.27 | 2.36 ± 0.92 |
| | | | *Pinus sylvestris* | Planted | 86 ± 3 | 328 ± 4 | 24 ± 3 | 27.00 ± 1.94 | 4.31 ± 1.28 |
| | | | *Picea abies* | Planted | 85 ± 4 | 446 ± 6 | 25 ± 5 | 27.50 ± 1.96 | 5.94 ± 1.81 |
| | | Gleyded | *Quercus robur* | Natural | 97 ± 0 | 241 ± 5 | 23 ± 2 | 28.50 ± 1.89 | 4.08 ± 1.20 |
| | | | *Pinus sylvestris* | Planted | 86 ± 1 | 334 ± 11 | 23 ± 0 | 27.00 ± 1.89 | 4.97 ± 1.28 |
| | | | *Picea abies* | Planted | 84 ± 1 | 376 ± 10 | 23 ± 3 | 25.00 ± 2.06 | 5.48 ± 2.21 |
| | Montane | Foothill | *Picea abies* | Planted | 75 ± 1 | 365 ± 30 | 24 ± 5 | 29.50 ± 2.82 | 8.18 ± 6.2 |
| | | Ravine | *Pinus sylvestris* | Natural | 123 ± 0 | 256 ± 6 | 22 ± 4 | 28.50 ± 2.50 | 3.39 ± 2.38 |
| | | Exposed | *Picea abies* | Planted | 96 ± 3 | 505 ± 16 | 28 ± 6 | 36.50 ± 3.65 | 5.89 ± 1.74 |
| | | Nutrient-rich | *Picea abies* | Planted | 76 ± 8 | 458 ± 37 | 26 ± 14 | 30.00 ± 3.58 | 6.55 ± 2.04 |
| | | Gleyed | *Picea abies* | Planted | 78 ± 6 | 391 ± 35 | 27 ± 9 | 33.00 ± 3.68 | 8.67 ± 7.56 |
| | | Wet | *Picea abies* | Planted | 63 ± 8 | 252 ± 22 | 20 ± 11 | 23.50 ± 3.81 | 6.52 ± 2.16 |
| | | Timberline | *Picea abies* | Natural | 90 ± 6 | 346 ± 15 | 21 ± 8 | 24.50 ± 4.70 | 4.03 ± 1.24 |
| Carpathians | Floodplain | Swamp | *Salix fragilis* | Natural | 45 ± 2 | 220 ± 10 | 14 ± 8 | 36.40 ± 2.04 | 2.02 ± 1.76 |
| | | Riverine | *Fraxinus excelsior* | Natural | 74 ± 3 | 402 ± 19 | 26 ± 6 | 31.50 ± 2.07 | 8.47 ± 2.87 |
| | Montane | Foothill | *Picea abies* | Planted | 79 ± 5 | 227 ± 12 | 28 ± 2 | 39.06 ± 2.77 | 7.13 ± 5.81 |
| | | Ravine | *Picea abies* | Planted | 72 ± 4 | 216 ± 13 | 18 ± 1 | 26.50 ± 4.76 | 3.45 ± 1.66 |
| | | Exposed | *Picea abies* | Planted | 89 ± 4 | 555 ± 22 | 30 ± 8 | 33.50 ± 3.27 | 7.14 ± 1.86 |
| | | Nutrient-rich | *Picea abies* | Planted | 79 ± 4 | 462 ± 15 | 26 ± 1 | 32.50 ± 3.71 | 7.32 ± 2.63 |
| | | Timberline | *Picea abies* | Natural | 86 ± 6 | 240 ± 20 | 20 ± 8 | 27.50 ± 4.98 | 3.25 ± 1.33 |

$V_{ha}$—volume stock (m$^3$/ha); h—height (m); $d_{1.3}$—diameter in breast height (cm); BAI—basal-area increment (m$^2$/ha).

### 3.3. Response of Forest Stands

The significant correlations of more than 50% of the growth condition properties qualified multiple regressions explaining > 50% of the BAI variance. However, the significance of multiple regressions was not reduced by more frequent insignificant simple correlations.

Temperatures, atmospheric $NO_x$ and $O_3$ concentrations and the soil clay content significantly affected the most evaluated stand types. Precipitation, temperatures, atmospheric $SO_2$, soil clay, pH, base saturation, $Al_2O_3$ and CaO impacted BAI in more than 75% of the stand types. On the contrary, the development of soil $C_{org}$ significantly influenced merely less than 12% of the compared stand types. Minor significant correlations involving < 50% of the properties were identified in 73% of the stand types while the predominant minor significant correlations of most ecosystem properties affected only 27% of the stand types. BAI of the planted rock pines, gleyed pine stands, nutrient-rich oak forests and upland gleyed spruce stands, as well as the wet montane spruce stands and Carpathian ravine forests, were influenced by the significant or low correlations covering more than 90% of the growth conditions. Significant correlations of more than 50% of the evaluated growth conditions impacted BAI not only in the Carpathian ravine forests but also in the East-Sudeten montane spruce forests, East-Sudeten gleyed spruce stands and the gleyed oak stands. The most influential growth conditions were components of acid deposition and soil physico-chemical pH and BS, partly also temperatures, precipitation, soil clay, $C_{org}$, $Al_2O_3$ and CaO (Table 5).

Multiple regressions among the BAI of the individual stand types and growth conditions included the range $r^2$ 0.16–0.92. The lowest multiple $r^2$ occurred in the upland exposed series. The highest $r^2$ was ascertained on ravine Carpathian slopes. The smallest difference between incremental $r^2$ within one stand type was determined at pine stands planted in rock cities. The greatest difference in $r^2$ was detected between the spruce stands in flat uplands. The site distributed multiple $r^2$ more distinctively than tree species. The Carpathian forests varied from the Eastern Sudetes by a greater $r^2$ BAI function within all soil series. The regressions of the Carpathian forests were more similar due to the positive parameters of $SO_2$ and soil pH, while East-Sudeten montane forests were dependent on the same parameters for atmospheric precipitation, $SO_2$ and soil pH, BS as well as $C_{org}$. In general, the residual normality was maintained in the conditions of the Bohemian Massif. The most similar residues occurred in pines, flat and exposed uplands, as well as in the montane wet and timberline sites.

On average, the highest values of $r^2 > 0.7$ of multiple regressions were detected in all the investigated montane forests and in the North Bohemian natural pines separately. The non-waterlogged submontane natural forests were characterised by the tightness of multiple regressions of $r^2 < 0.6$, but the nutrient-rich forests reached an $r^2$ between 0.60–0.77. The naturally upland oak stands evinced the most significant regression of BAI at the optimal nutrient-rich sites. By contrast, the spruce stands were less dependent on the condition development than the pine and oak ones in all the intra-Bohemian soil series. The higher dependence in natural upland forests was conditioned by differentiated parameters at atmospheric properties and by equally directional parameters of soil clay, pH, BS, $C_{org}$ and CaO (Table 6).

**Table 5.** Linear correlations and analysis of residues between tree current basal-area increment and properties of growth conditions at sampled forest stand type. **Bold** statistically significant correlation at $p < 0.05$, normal less significant between $p \geq 0.05$ and $p < 0.50$ and light insignificant at $p \geq 0.50$. *RSD*—residual standard deviation; $E_R$—residual elevation; $A_R$—residual skewness (for detailed information about properties of growth conditions see Tables 2 and 3).

| Geotectonics | Relief | Series | Tree-Species | Origin | *T* | *P* | SO$_2$ | NO$_x$ | O$_3$ | clay | pH | BS | C$_{org}$ | Al$_2$O$_3$ | CaO | *RSD* | $E_R$ | $A_R$ |
|---|---|---|---|---|---|---|---|---|---|---|---|---|---|---|---|---|---|---|
| Bohemian Massif | Rocky | Pine | *Pinus sylvestris* | Natural | −0.09 | 0.02 | **0.39** | 0.08 | −0.03 | **0.29** | 0.16 | −0.08 | **−0.29** | 0.18 | −0.08 | 0.24 | **−0.14** | **−0.72** |
| | | | *Pinus sylvestris* | Planted | 0.12 | −0.12 | **0.29** | 0.20 | **−0.28** | 0.10 | −0.08 | −0.10 | −0.13 | 0.13 | −0.02 | 0.24 | **−0.02** | 0.39 |
| | Upland | Flat | *Picea abies* | Planted | −0.01 | 0.23 | 0.07 | −0.21 | 0.15 | **0.24** | 0.24 | 0.22 | −0.14 | −0.01 | 0.12 | 1.37 | **−1.44** | **0.05** |
| | | Exposed | *Quercus petraea* | Natural | 0.10 | −0.17 | −0.03 | 0.01 | −0.11 | 0.00 | **−0.32** | −0.10 | 0.06 | 0.04 | **−0.28** | 0.30 | **1.81** | 1.29 |
| | | | *Pinus sylvestris* | Planted | −0.28 | 0.07 | 0.09 | 0.23 | −0.07 | 0.03 | 0.07 | 0.05 | −0.04 | 0.16 | −0.03 | 0.21 | **−0.47** | **−0.26** |
| | | | *Picea abies* | Planted | −0.26 | 0.17 | 0.13 | −0.41 | 0.19 | −0.05 | 0.02 | −0.11 | −0.06 | −0.01 | −0.08 | 8.51 | **−0.50** | **−0.28** |
| | | Nutrient−rich | *Quercus petraea* | Natural | 0.11 | 0.09 | −0.18 | **−0.31** | **0.32** | −0.16 | **−0.36** | −0.10 | 0.06 | −0.09 | −0.18 | 0.22 | 1.22 | 0.88 |
| | | | *Pinus sylvestris* | Planted | **−0.36** | −0.11 | 0.01 | **0.51** | **−0.34** | **0.35** | 0.17 | −0.04 | −0.15 | 0.18 | 0.12 | 0.24 | **−0.13** | **−0.18** |
| | | | *Picea abies* | Planted | −0.05 | −0.05 | −0.12 | **0.27** | −0.15 | **0.27** | 0.17 | −0.01 | −0.08 | 0.00 | 0.11 | 0.43 | **−0.31** | 0.16 |
| | | Gleyded | *Quercus robur* | Natural | −0.09 | **−0.29** | **0.38** | **0.34** | **−0.29** | 0.04 | −0.12 | **−0.27** | 0.06 | −0.13 | **−0.24** | 0.21 | **−0.76** | 0.22 |
| | | | *Pinus sylvestris* | Planted | 0.00 | −0.01 | **−0.23** | **0.19** | −0.07 | **0.39** | **0.23** | −0.17 | −0.09 | **0.30** | 0.02 | 0.19 | **−0.26** | 0.32 |
| | | | *Picea abies* | Planted | −0.12 | −0.12 | 0.10 | **0.41** | **−0.35** | 0.13 | 0.12 | −0.12 | 0.09 | 0.00 | −0.13 | 0.45 | 0.70 | 0.99 |

**Table 5.** *Cont.*

| Geotectonics | Relief | Series | Tree-Species | Origin | T | P | SO₂ | NOₓ | O₃ | clay | pH | BS | Corg | Al₂O₃ | CaO | RSD | ER | AR |
|---|---|---|---|---|---|---|---|---|---|---|---|---|---|---|---|---|---|---|
| | | Foothill | *Picea abies* | Planted | **0.51** | **−0.30** | 0.10 | 0.10 | **−0.67** | −0.09 | 0.03 | 0.00 | 0.07 | 0.00 | −0.01 | 0.98 | **−0.64** | **0.45** |
| | | Ravine | *Pinus sylvestris* | Natural | 0.07 | 0.10 | **0.32** | −0.03 | −0.06 | 0.10 | 0.00 | 0.02 | −0.03 | 0.13 | 0.10 | 0.17 | 0.43 | 0.03 |
| | | Exposed | *Picea abies* | Planted | −0.03 | −0.01 | **−0.30** | 0.04 | 0.00 | −0.06 | 0.05 | 0.10 | 0.13 | −0.14 | 0.03 | 0.28 | 1.36 | 0.18 |
| | Montane | Nutrient−rich | *Picea abies* | Planted | **0.29** | **−0.26** | 0.16 | −0.07 | **0.55** | 0.08 | −0.18 | −0.10 | −0.01 | 0.16 | 0.14 | 0.89 | 4.38 | 1.28 |
| | | Gleyed | *Picea abies* | Planted | **−0.22** | −0.12 | **0.27** | **0.40** | **−0.45** | **−0.25** | 0.21 | **0.33** | −0.07 | 0.12 | 0.00 | 0.92 | 1.60 | **−0.81** |
| | | Wet | *Picea abies* | Planted | −0.06 | −0.14 | 0.18 | **0.31** | **−0.29** | −0.17 | **0.38** | **0.48** | −0.13 | **0.24** | −0.11 | 0.54 | **−0.81** | **−0.22** |
| | | Timberline | *Picea abies* | Natural | 0.02 | **−0.43** | 0.06 | **0.28** | **−0.38** | **0.40** | **0.33** | 0.16 | **0.33** | −0.19 | −0.01 | 0.42 | **−0.79** | **−0.44** |
| Carpathians | Floodplain | Swamp | *Salix fragilis* | Natural | 0.14 | 0.20 | −0.06 | **−0.23** | 0.03 | **−0.24** | −0.03 | 0.06 | 0.03 | **−0.24** | −0.04 | 1.03 | 3.31 | **−0.02** |
| | | Riverine | *Fraxinus excelsior* | Natural | **−0.32** | −0.15 | **0.24** | 0.02 | 0.19 | **0.31** | 0.16 | −0.19 | **−0.23** | **0.25** | −0.02 | 0.71 | 2.18 | 1.64 |
| | | Foothill | *Picea abies* | Planted | **−0.29** | **−0.28** | 0.20 | 0.03 | −0.01 | **−0.21** | 0.05 | −0.10 | −0.18 | **0.31** | 0.01 | 0.52 | 4.06 | 2.00 |
| | | Ravine | *Picea abies* | Planted | **0.27** | −0.13 | **−0.36** | −0.09 | −0.14 | 0.19 | **0.37** | **0.38** | −0.02 | **0.42** | **0.35** | 0.25 | 1.37 | 1.33 |
| | Montane | Exposed | *Picea abies* | Planted | **0.42** | **−0.40** | −0.01 | **−0.46** | **−0.39** | 0.07 | **0.30** | 0.14 | −0.06 | **0.22** | **0.25** | 0.68 | 2.81 | **−0.03** |
| | | Nutrient−rich | *Picea abies* | Planted | 0.14 | **−0.49** | 0.13 | 0.00 | **−0.43** | 0.03 | **0.43** | 0.14 | −0.11 | **0.32** | **0.33** | 0.33 | 0.16 | 0.45 |
| | | Timberline | *Picea abies* | Natural | 0.03 | −0.08 | −0.17 | 0.08 | 0.05 | 0.10 | **0.26** | **0.32** | −0.01 | **0.35** | **0.36** | 0.38 | 2.35 | 1.46 |

**Table 6.** Multiple regressions between current basal-area increment and properties of growth conditions at forest stand types (for detail information about properties of growth conditions see Tables 2 and 3).

| Geotectonics | Relief | Series | Tree-Species | Origin | $F_{min}$ | $r^2$ | T | P | SO$_2$ | NO$_x$ | O$_3$ | clay | pH | BS | C$_{org}$ | Al$_2$O$_3$ | CaO | b |
|---|---|---|---|---|---|---|---|---|---|---|---|---|---|---|---|---|---|---|
| Bohemian Massif | Rocky | Pine | *Pinus sylvestris* | Natural | 5.07 | 0.61–0.85 | 0.341 | −0.179 | 0.552 | −0.587 | −0.546 | 0.248 | −0.049 | 0.469 | −0.041 | 0.300 | −0.521 | 4.406 |
| | | | *Pinus sylvestris* | Planted | 1.58 | 0.40–0.45 | 0.140 | 0.174 | 0.544 | −0.081 | −0.059 | −0.021 | −0.025 | −0.080 | −0.050 | 0.155 | 0.272 | 3.017 |
| | Upland | Flat | *Picea abies* | Planted | 0.57 | 0.20–0.64 | 0.360 | 1.974 | 5.069 | −2.422 | 1.813 | −0.537 | −0.552 | −0.153 | −1.830 | −1.498 | 0.385 | 10.233 |
| | | Exposed | *Quercus petraea* | Natural | 0.83 | 0.26–0.55 | −0.148 | 0.028 | 0.118 | −0.109 | 0.227 | −0.018 | −0.242 | 0.131 | −0.030 | 0.554 | −0.549 | 3.097 |
| | | | *Pinus sylvestris* | Planted | 1.86 | 0.44–0.69 | 0.060 | −0.050 | −0.395 | 1.468 | 0.104 | −0.164 | −0.097 | 0.248 | −0.150 | 0.479 | −0.642 | 4.400 |
| | | | *Picea abies* | Planted | 0.44 | 0.16–0.43 | −0.058 | −0.209 | −0.105 | −0.249 | −0.055 | −0.141 | −0.672 | 0.862 | −0.064 | −0.045 | −0.160 | 4.625 |
| | | Nutrient−rich | *Quercus petraea* | Natural | 1.59 | 0.40–0.70 | −0.079 | 0.198 | −0.704 | 0.199 | 0.134 | −0.347 | −0.520 | 0.514 | −0.293 | −0.158 | −0.293 | 2.814 |
| | | | *Pinus sylvestris* | Planted | 4.13 | 0.64–0.71 | −0.133 | −0.022 | −0.526 | 1.116 | 0.090 | 0.006 | −0.077 | −0.040 | 0.047 | 0.383 | −0.457 | 4.865 |
| | | | *Picea abies* | Planted | 2.06 | 0.47–0.58 | 0.590 | −0.001 | 0.507 | −0.710 | −1.229 | −0.717 | −0.278 | 0.718 | −0.149 | 0.595 | −0.723 | 6.887 |
| | | Gleyded | *Quercus robur* | Natural | 2.91 | 0.55–0.64 | 0.136 | −0.161 | 0.238 | 0.229 | −0.148 | −0.272 | −0.235 | 0.072 | −0.030 | 0.177 | −0.102 | 4.125 |
| | | | *Pinus sylvestris* | Planted | 3.52 | 0.47–0.55 | 0.365 | −0.026 | −0.323 | 0.178 | −0.111 | −0.473 | −0.405 | 0.117 | 0.400 | 0.236 | 0.364 | 5.810 |
| | | | *Picea abies* | Planted | 2.96 | 0.40–0.48 | −0.304 | −0.441 | −0.061 | −0.944 | −0.304 | −0.770 | −1.458 | 0.431 | 0.225 | 1.412 | 0.706 | 4.887 |
| | Montane | Foothill | *Picea abies* | Planted | 1.17 | 0.33–0.50 | −0.206 | 0.243 | 0.154 | −0.834 | −0.544 | −1.233 | −1.350 | 0.037 | 0.045 | 0.921 | −0.088 | 9.057 |
| | | Ravine | *Pinus sylvestris* | Natural | 3.04 | 0.43–0.52 | −0.025 | 0.152 | 0.409 | 0.351 | −0.063 | 0.911 | −0.181 | −0.545 | 0.299 | 0.179 | 0.442 | 2.858 |
| | | Exposed | *Picea abies* | Planted | 4.23 | 0.56–0.72 | 0.382 | −0.223 | −0.090 | −0.131 | 0.457 | −0.447 | −0.556 | 0.025 | −0.682 | −0.717 | 0.610 | 6.117 |
| | | Nutrient-rich | *Picea abies* | Planted | 0.98 | 0.23–0.42 | −0.301 | −0.444 | −0.292 | −0.081 | −0.653 | −0.603 | −0.628 | 0.508 | −0.275 | −0.441 | 0.791 | 6.777 |
| | | Gleyed | *Picea abies* | Planted | 3.99 | 0.55–0.85 | 2.567 | −0.051 | −8.886 | 0.066 | −1.305 | 0.472 | −25.071 | 23.376 | −14.938 | 4.943 | 4.655 | 10.945 |
| | | Wet | *Picea abies* | Planted | 6.45 | 0.70–0.76 | 1.278 | −3.053 | −14.224 | 19.046 | 21.629 | 4.129 | −18.899 | 12.463 | −18.377 | 19.398 | −3.738 | 19.110 |
| | | Timberline | *Picea abies* | Natural | 11.60 | 0.78–0.84 | 0.019 | −0.141 | −0.583 | −0.631 | 0.795 | −0.381 | −0.392 | 0.405 | −0.597 | −0.275 | 0.046 | 3.875 |

**Table 6.** *Cont.*

| Geotectonics | Relief | Series | Tree-Species | Origin | $F_{min}$ | $r^2$ | $T$ | $P$ | $SO_2$ | $NO_x$ | $O_3$ | clay | pH | BS | $C_{org}$ | $Al_2O_3$ | CaO | $b$ |
|---|---|---|---|---|---|---|---|---|---|---|---|---|---|---|---|---|---|---|
| Carpathians | Floodplain | Swamp | *Salix fragilis* | Natural | 2.69 | 0.53–0.60 | 0.245 | −0.185 | 1.048 | 0.336 | 0.074 | 2.598 | 0.683 | 0.000 | 1.107 | −1.858 | −0.363 | 1.758 |
| | | Riverine | *Fraxinus excelsior* | Natural | 1.63 | 0.41–0.77 | −0.504 | 1.055 | −0.860 | 0.713 | −0.024 | −2.397 | 1.598 | −0.888 | −1.602 | 0.808 | −0.518 | 8.951 |
| | Montane | Foothill | *Picea abies* | Planted | 4.11 | 0.59–0.66 | 2.635 | −0.003 | 0.226 | 1.251 | 0.025 | 0.562 | −18.671 | 0.033 | 0.585 | 0.073 | 0.640 | 41.357 |
| | | Ravine | *Picea abies* | Planted | 14.89 | 0.82–0.92 | −0.303 | 1.254 | 0.243 | 0.844 | −0.148 | 0.081 | 0.305 | −0.071 | 0.798 | 0.652 | −0.704 | 3.698 |
| | | Exposed | *Picea abies* | Planted | 8.71 | 0.73–0.84 | −0.015 | 0.116 | −0.862 | −0.292 | 0.122 | −0.290 | 0.132 | 0.021 | −0.230 | 0.303 | −0.402 | 7.097 |
| | | Nutrient−rich | *Picea abies* | Planted | 5.86 | 0.64–0.70 | −0.644 | 0.793 | 0.976 | −1.067 | 0.326 | 0.591 | −0.361 | −0.793 | 0.245 | 0.850 | −0.027 | 8.381 |
| | | Timberline | *Picea abies* | Natural | 6.12 | 0.72–0.91 | −0.032 | −0.197 | −0.462 | 0.113 | −0.200 | 0.209 | 0.604 | 0.689 | −0.872 | −0.748 | 0.864 | 4.013 |

## 4. Discussion

Multiple dependence of the current radial increment on external conditions was most significantly distributed along different sites. On the other hand, the natural forest stands are more closely reflected by the effects of changing site conditions than unnatural tree species populations. The dependence proximity indicated the magnitude of the forest tree species increment response to multiple variabilities of growth conditions.

The forest tree species increment naturally depends on environmental properties as well as on the competition of individual species. The assessment of different tree species' responses has confirmed diversification by fertility, regional climate change and atmospheric pollution [3,24,29]. Environment significantly influenced the increment of most of the compared forest stand types. However, the parameters of individual environmental properties differed in size and direction among stand types. Atmospheric precipitation, soil clay, pH, $C_{org}$ and $Al_2O_3$ influenced most forest stands to the same extent. Tropospheric $O_3$ and soil $C_{org}$ and $Al_2O_3$ significantly affected forests in the individual areas contradictorily. The opposing effect is related to the geographical location, stimulating diverse impacts of regional climate change together with age and forest stand density [56].

Air pollution has reduced tree species increments, particularly in mountain altitudes. Nonetheless, the individual pollutants acted opposite among the different nutrient-rich sites from the uplands to the mountains. Atmospheric $SO_2$ has significantly affected the natural pine forests, intra-Bohemian gleyed forests, and East-Sudeten ravine to exposed ecosystems, floodplain ash stands and Carpathian ravine forests. $NO_x$ pollution has affected several monitored stands from the natural pines, nutrient-rich and gleyed upland forests and waterlogged East-Sudeten ecosystems to the timberline. In the Outer-Carpathian areas, the willow floodplains and exposed slopes were particularly impacted. Tropospheric $O_3$ significantly influenced (especially) nutrient-rich to gleyed upland communities in the Eastern Sudetes. The increment of the Carpathian forests was most impacted by tropospheric $O_3$ concentrations in the nutrient-rich exposed sites. The development of $SO_2$ concentrations reduced forest increment most significantly in the gleyed and exposed sites from the uplands to the mountains while the $NO_x$ course contributed to the forest increment increase in lower altitudes and reduced the increment of montane forests. Contrarily, $O_3$ in the lower altitudes mostly reduced forest increment but did not affect selected montane forests markedly.

The adverse effects of atmospheric pollution are associated with plant–climate interactions. The temperature development mostly influences montane forest growth, whereas upland forests depend more on available soil water [57]. The climate change impacts are modified by the complicated arrangement of mountain relief. Relief contributes to forest vulnerability with damage exceeding 32% of the tree species population. Forests growing at altitudes $\leq$ 950 m a.s.l. are affected more by drought than by pollution but the forest development > 950 m a.s.l. was impacted by the long-distance transmission of pollution and their increment was increased after the decrease in acidic load due to global warming [58]. The altitude exceeds 90% of the relief risk while the slope, exposure and relief shape include only less than 7% of the predispositions. Montane forest damage is mainly concentrated on ridges, higher slope parts, round valleys and northern to north-western exposures due to the relief effects [59]. Simultaneously, the climate change impacts on forests are more distinctive in mountain locations than in the uplands. On the one hand, climate change increases tree species increment and prolongs the growth dynamics of older stands. On the other hand, it increases the frequency of sudden weather disturbances [4,25,43].

On the one hand, the close relationship between tree increment and environmental properties indicated compliance with the site, but on the other hand, it indicated sensitivity to the change in any environmental properties. The increment of the nutrient-rich upland forests was dependent on environmental variability more than on the mountain altitudes. The unnatural pine stands in the upland locations differed slightly in their dependence on the environmental properties due to natural oak groves. Nevertheless, the pine stands grown in rock cities were significantly less dependent on the environment than natural

pines. Scots pine appears to be very sensitive to growth condition variability as it is most sensitive to drought from the Central-European managementally important tree species [6]. Even though Norway spruce or European beech may also be susceptible to drought, the extended growing season supported their growth in the humid montane altitudes [29]. While the growth of the natural pines followed fluctuating atmospheric precipitation and higher $O_3$ load, the oak stands under natural conditions reacted by means of an opposite decrease in their dependence with a more variable environment on exposed sites and by an increase on nutrient-rich sites. The influence of nutrient-rich sites on tree species increment decreased with the altitude. On the contrary, the increment of upland gleyed forests was contingent on the environment less than in mountain altitudes. The waterlogged oak groves, including floodplain forests, appear to be more resistant to environmental change in upland altitudes due to a sufficient supply of soil water to the sites [60].

## 5. Conclusions

The atmospheric and soil properties influenced diameter increment development of Central-European forest tree species differentially among the montane, upland and waterlogged sites. Temperatures, acid deposition and soil physico-chemical properties, $Al_2O_3$ and $CaO$ significantly affected more than 75% of the stand types. The forests adaptable to variable growth conditions were characteristic by multiple regression $r^2 \geq 0.60$. Spruce and pine stands growing in upland altitudes seemed to be more susceptible to growth condition variability than natural forests. Resistant stands were defined by natural forests at sites sufficiently supplied with water. Susceptibility to the environmental change was higher at higher altitudes. High-mountain forests, as well as the natural pines, seemed to be one of the most sensitive species to environmental change.

**Author Contributions:** Conceptualization, P.S.; methodology, P.S., M.Z. (Miroslav Zeman) and M.Z. (Miloš Zapletal); validation, P.R. and M.Z. (Miroslav Zeman); formal analysis, P.S. and P.R.; investigation, P.S. and P.R.; resources, P.S.; data curation, P.R.; writing—original draft preparation, P.S.; writing—review and editing, P.S. and M.Z. (Miloš Zapletal).; visualization, P.S.; supervision, M.Z. (Miloš Zapletal). All authors have read and agreed to the published version of the manuscript.

**Funding:** The study has received funding from the European Union's Horizon 2020 Programme for Research & Innovation under grant agreement No 952314 ASFORCLIC.

**Data Availability Statement:** Restrictions apply to the availability of data used. Data was obtained from third party and are available from the authors with the permission of data owner.

**Conflicts of Interest:** The authors declare no conflict of interest.

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
