# Peer review of "Environmental Effects among Differently Located and Fertile Sites on Forest Basal-Area Increment in Temperate Zone"

_forests, doi:10.3390/f13040588_

Round 1

Author Response

Reviewer’s 1 Comment #1: A last paragraph (of the Introduction) is missing: with general and specific aims and objectives; and may also raise some hypotheses.

The Authors Response to the Reviewer’s 1 Comment #1: Thank You for the comment. The aim of the work was specified in a paragraph above. We considered that the aim specification in two paragraphs would be better for explanation. Thus, the penultimate introduction paragraph was focused on main goal of the work including working hypothesis, while last introduction paragraph was focused on princiles of approach used for answers to working hypotheses. The goals were defined as main by assessment of forest response on environmental change among different sites, and as secondary by the assessment of the response on differently forcing environmental properties. The working hypotheses were specified as presumptions about parameters suggesting forest resilience and about influence of distribution as well as structure of cultural forests on assessment quality. Consequently, the last introduction paragraph was used for a brief description of approach robust under limits due to forest distribution in cultural landscape.

Reviewer’s 1 Comment #2: Should start with the characterization of the study area and only after starting to describe the geographic division.

The Authors Response to the Reviewer’s 1 Comment #2: We apologize that the section title could to confuse You. The subtitle “Geographical division” was corrected to “Geographical pattern” which could inform about development of approaches to solution of the working hypotheses defined in last two introduction paragraphs. The Geographical pattern section describes situation conditional for a plot selection as well as site and stand general characteristics used for the selection.

Reviewer’s 1 Comment #3: Start "Field Sampling" with this part and describe how many plots were sampled.

The Authors Response to the Reviewer’s 1 Comment #3: Here, we input brief information about 52 selected plots.

Reviewer’s 1 Comment #4: You can include Statistical Analyzes as a separate subtitle.

The Authors Response to the Reviewer’s 1 Comment #4: The methodical paragraphs about statistical analyses were separated into unique subsection 2.4. This separation required simple correction in first sentence of the original section 2.3, and (contrarily) also simple specification on multiple regression through adding information p < 0.05.

Reviewer’s 1 Comment #5: Improve and develop the discussion. Perhaps it will be more noticeable if you reorganize the discussion according to the subtitles of the results.

The Authors Response to the Reviewer’s 1 Comment #5: We carried out reorganization at sentence and discussion paragraph ordering. The reorganization was concentrated to discussion in sequence similar with particular results. First, the results about effects of all environmental properties were discussed, second, the results about harmful pollution were discussed, third, the effects from altitudinal location and relief were discussed together in one paragraph, and finally, synthesis was derived.

Reviewer 2 Report

The manuscript: “Multiple Effects of Variable Environmental Properties on Basal-Area Increment in Forests of Various Tree-Species among  Differently Located and Fertile Sites under Temperate Conditions” deals with an important problem – assessing the impact of different factors on the BAI of different tree species. The value of the analyses is all the higher because of the use of a relatively long time series of analyzed data in the period 1971- 2018 and analyzed a wide set of factors shaping the growth of tree diameter (basal area).

However, the manuscript needs some improvements.

Consider changing/simplifying the title - it contains too much information and thus does not have a clear, simple message and is very difficult to read.

1 line 35 and 37 repetition

  1. line 198 – p<0.5 seems not significant – consider the slightly increased significance.
  2. line 206 “where xi is the z-transformed” – shortly explain “z-transformation.
  3. line 289 “The lowest increment” it should be “The lowest BAI increment”, 5 > 8 m2 /ha.year – why dot is used –(by the way, it seem very high).

Line 310 is  r2 correct? consider changing on capital letter R2

Table 1. symbols (A, B, …) should be explained in the footer of the table.

Table 2, Table 3 - remove the description of symbols from the title to the footer of the table, explain the value “±” is it standard deviation?

Table 4.  “…d1.3 – diameter in breast height” give unit (cm)

; BAI – basal-area increment 432 (m3 /ha). It should be m2

Author Response

Reviewer’s 2 Comment #1: Consider changing/simplifying the title - it contains too much information and thus does not have a clear, simple message and is very difficult to read.

The Authors Response to the Reviewer’s 2 Comment #1: The title was simplified to “Environmental Effects among Differently Located and Fertile Sites on Forest Basal-Area Increment in Temperate Zone”.

Reviewer’s 2 Comment #2: 1 line 35 and 37 repetition

The Authors Response to the Reviewer’s 2 Comment #2: The repeat chapter title “1 Introduction” was reduced to one place.

Reviewer’s 2 Comment #3: line 198 – p<0.5 seems not significant – consider the slightly increased significance.

The Authors Response to the Reviewer’s 2 Comment #3: Thank You for the recommendation. We would like to write that we understand Your apprehensions. But we decided to keep descripted approach according to general theory of probability. The interval between p≥0.05 and p<0.50 included wide range of significance levels from relatively accurate characterizing about 90% or 75% of total data variability to edge probable near 51%. This situation is similar with criterions of logistic regression, where hypothesis is recognizable under probability higher than 50%. But for our environmental analysis, we rather used linear regression approximating relationships between real variables.

Reviewer’s 2 Comment #4: line 206 “where xi is the z-transformed” – shortly explain “z-transformation.

The Authors Response to the Reviewer’s 2 Comment #4: The z-transformation was explained by sentence: “Independent environmental variables were normalized by z-transformation expressing deviations of the measured values from mean.”

Reviewer’s 2 Comment #5: line 289 “The lowest increment” it should be “The lowest BAI increment”, 5 > 8 m2 /ha.year – why dot is used –(by the way, it seem very high).

The Authors Response to the Reviewer’s 2 Comment #5: Thank You for the recommendation. Of course, the term “increment” was specified as basal-area increment or by abbreviation BAI and we added short information that high values of the BAIs were obtained on rich floodplain or foothill sites.

Reviewer’s 2 Comment #6: Line 310 is  r2 correct? consider changing on capital letter R2

The Authors Response to the Reviewer’s 2 Comment #6: Yes, we consider the abbreviation r2 right like we defined it as determination index.

Reviewer’s 2 Comment #7: Table 1. symbols (A, B, …) should be explained in the footer of the table.

The Authors Response to the Reviewer’s 2 Comment #7: Symbols A, B, C … were explained under Figure 1 with mentioned reference in Table 1.

Reviewer’s 2 Comment #8: Table 2, Table 3 - remove the description of symbols from the title to the footer of the table, explain the value “±” is it standard deviation?

The Authors Response to the Reviewer’s 2 Comment #8: The descriptions of the symbols were moved to footers under all tables.

Reviewer’s 2 Comment #9: Table 4.  “…d1.3 – diameter in breast height” give unit (cm)

The Authors Response to the Reviewer’s 2 Comment #9: The unit (cm) was added.

Reviewer’s 2 Comment #10: BAI – basal-area increment 432 (m3 /ha). It should be m2

The Authors Response to the Reviewer’s 2 Comment #10: The m3 were corrected to m2 in relation to stand basal-area.

Reviewer 3 Report

General remarks of the reviewer

Title: The title is too extensive, it can be changed to a more condensed one. Details are included in the keywords and purpose of the work.

Abstract: The abstract gives a good overview of the work.

Keywords: The keywords are specific to the topic under study.

Introduction: The state of the research  is reviewed  and major publications cited.

Materials and Methods: The empirical material is sufficient in terms of quantity and quality. Appropriate research methodology and proper statistical analysis of the data were applied.

Results: The results were correctly analyzed statistically and presented well in tables. Please link them in the text with figures 2,3,4.

Discussion: The research results are well-discussed.

Conclusions: The conclusions are constructive.

References: For publications, please add DOI.

Technical Notes

Please correct any minor inaccuracies in the text of the article.

The description of the literature item needs to be corrected as required by the publisher: articles, books and other sources - italics of journal titles, year in bold, correct pages of journals and the access link and date of access in English. According to MDPI standard.

Details in the attached manuscript.

Summary of the review:

The article exhausts the presented issue.

Author Response

Reviewer’s 3 Comment #1: The title is too extensive, it can be changed to a more condensed one. Details are included in the keywords and purpose of the work.

The Authors Response to the Reviewer’s 3 Comment #1: The title was simplified. Please, see to our response on the Reviewer’s 2 Comment #1.

Reviewer’s 3 Comment #2: The results were correctly analyzed statistically and presented well in tables. Please link them in the text with figures 2,3,4.

The Authors Response to the Reviewer’s 3 Comment #2: Thank You for the recommendation. The links belonging to figures 2 and 3 were added into Material and Method section because they characterized input data, while link to the figure 4 was added to results about stand characteristics to demonstration of assessed dependent variables.

Reviewer’s 3 Comment #3: For publications, please add DOI. The description of the literature item needs to be corrected as required by the publisher: articles, books and other sources - italics of journal titles, year in bold, correct pages of journals and the access link and date of access in English. According to MDPI standard.

The Authors Response to the Reviewer’s 3 Comment #3: We would like to thank You for the recommendation. We controlled all references and compared them with Instructions for Authors as well as with examples in current articles. Unfortunately, we could not see DOI code anywhere. But we are ready to correct all references needed according to direct instructions from the Editor.

Round 2

Reviewer 1 Report

Please see comments in the attached file.

Author Response

Dear Sirs,

Thank you and the Reviewers for the careful reading and constructive recommendations for improving our manuscript. All changes in the text were done under regime Track Changes. Here, we would like to present our responses to particular recommendations from the reviewer.

Reviewer’s 1 Comment #1: The paragraph with the objects should be the last paragraph of the introduction.

The Authors Response to the Reviewer’s 1 Comment #1: The characteristics of aims and working hypothesis of our study were concentrated into one paragraph placed as last at the Introduction chapter. By this, we preserved characteristics of the problematics within suggested approaches for investigation.

Reviewer’s 1 Comment #2: (lines 202-203) This description is not necessary , it is sufficient and it is implicit in the p<0.05. Please improve this sentence.

The Authors Response to the Reviewer’s 1 Comment #2: We consider about original sentence version because it was result from previous discussion with another reviewer. Thus we apologize for disappointment that our answer is not as You expect. The used evaluation of correlations as exploratory technics was based on general theory of probability, where the interval between p≥0.05 and p<0.50 included wide range of significance levels from relatively accurate characterizing about 90% or 75% of total data variability to edge probability near 51%. This situation is similar with criterions of logistic regression, where hypothesis is recognizable under probability higher than 50%. But for our environmental analysis, we rather used linear correlation approximating relationships between real variables. Anyway, we put description on assessment of linear correlations into captures belonging to Table 5.